# The Effects of Trauma, with or without PTSD, on the Transgenerational DNA Methylation Alterations in Human Offsprings

**DOI:** 10.3390/brainsci8050083

**Published:** 2018-05-08

**Authors:** Nagy A. Youssef, Laura Lockwood, Shaoyong Su, Guang Hao, Bart P. F. Rutten

**Affiliations:** 1Department of Psychiatry & Health Behavior, Medical College of Georgia at Augusta University, Augusta, GA 30912, USA; 2Academic Affairs, Medical College of Georgia, Augusta University, Augusta, GA 30912, USA; 3Department of Psychiatry and Neuropsychology, University of Alabama at Birmingham, Birmingham, AL 35233, USA; llockwood@uabmc.edu; 4Department of Population Health Sciences, Medical College of Georgia, Augusta University, Augusta, GA 30912, USA; SSU@augusta.edu (S.S.); GHAO@augusta.edu (G.H.); 5Department of Psychiatry and Neuropsychology, Maastricht University Medical Centre, School for Mental Health and Neuroscience, 6202 AZ Maastricht, The Netherlands; b.rutten@maastrichtuniversity.nl

**Keywords:** psychological trauma, post-traumatic stress disorder, epigenomics, DNA methylation, prevention, treatment, transgenerational effect, transgenerational inheritance

## Abstract

Exposure to psychological trauma is a strong risk factor for several debilitating disorders including post-traumatic stress disorder (PTSD) and depression. Besides the impact on mental well-being and behavior in the exposed individuals, it has been suggested that psychological trauma can affect the biology of the individuals, and even have biological and behavioral consequences on the offspring of exposed individuals. While knowledge of possible epigenetic underpinnings of the association between exposure to trauma and risk of PTSD has been discussed in several reviews, it remains to be established whether trauma-induced epigenetic modifications can be passed from traumatized individuals to subsequent generations of offspring. The aim of this paper is to review the emerging literature on evidence of transgenerational inheritance due to trauma exposure on the epigenetic mechanism of DNA methylation in humans. Our review found an accumulating amount of evidence of an enduring effect of trauma exposure to be passed to offspring transgenerationally via the epigenetic inheritance mechanism of DNA methylation alterations and has the capacity to change the expression of genes and the metabolome. This manuscript summarizes and critically reviews the relevant original human studies in this area. Thus, it provides an overview of where we stand, and a clearer vision of where we should go in terms of future research directions.

## 1. Introduction

Post-traumatic stress disorder (PTSD) is characterized by four symptom clusters as defined by the Diagnostic and Statistical Manual, Fifth Edition (DSM-5) [1]: re-experiencing, avoidance of stimuli associated with the trauma, negative cognitions and affect associated with the trauma, and hyperarousal symptoms and signs. Fifty to 85% of Americans experience at least one traumatic event during their lifetimes, but only 7.8% go on to develop PTSD [2]. Thus, the question that frequently arises is: why do some people develop PTSD after experiencing trauma, while others do not? Possible explanations could be: an underlying genetic or epigenetic risk in those who are more prone to develop PTSD and/or a protective (epi)genetic makeup, or some form of high psychological resiliency in those who do not develop PTSD or who quickly recover from PTSD, i.e., resilient individuals.

PTSD has long been established to be due to exposure to trauma, and it has been assumed that only environmental factors would contribute to the development of PTSD. On the other hand, heritability of PTSD has been estimated to be between 30% and 70% in twin studies [3,4,5,6,7]. Genetic studies of both candidate gene and genome-wide association studies (GWAS) have provided many interesting and promising findings, yet (so far) no robust genetic variants for PTSD have been identified [8]. This risk is unlikely to be fully explained by only structural genetics [9].

Thus, the field of epigenetics could offer insights into differential susceptibility of risk to develop psychopathology. According to Goldberg et al. [9], epigenetics is the study of functionally stable and ideally/heritable changes in gene expression or cellular phenotype that occurs without changes in base pairing. In other words, epigenetics involves functional changes to the gene without sequence changes. The best studied epigenetic mechanism so far is via DNA methylation. We define DNA methylation as the attachment of methyl groups to the DNA molecule. When methyl groups are attached to the promoter, they typically act to repress gene transcription.

DNA methylation changes in genes have been associated with several psychiatric dysfunctions. For instance, we previously found an association between DNA methylation and both depression and suicide [10]. Hypermethylation of BDNF promoter or TrkB were especially involved in suicide [10]. Another equally interesting area of research is the epigenetic changes associated with PTSD.

Only a few reviews have rigorously examined this area of the literature. For instance, a review by Ramo-Fernandez et al. [11] focused primarily on the epigenetic changes associated with war trauma and childhood maltreatment. Also, a helpful review by Vinkers et al. discussed the effect of trauma on DNA methylation changes in humans, and categorized the studies reviewed depending on the timing of trauma exposure throughout the life span [12]. The authors pointed out “there are significant drawbacks in the existing human literature” including “lack of longitudinal studies, methodological heterogeneity, selection of tissue type, and the influence of developmental stage and trauma type on methylation outcomes” [12]. A recent longitudinal study by Rutten et al. examined the genome-wide blood DNA methylation profile changes as associated with the development of PTSD symptoms over time in two military cohorts (discovery, *n* = 93, and replication data sets, *n* = 98) [13]. The researchers found that development of PTSD symptoms over time in combat soldiers was significantly associated with DNA methylation changes [13]. The researchers suggested that the DNA methylation mediated the relation between combat trauma and PTSD symptoms longitudinally [13].

However, another equally important, but less explored area, that has not been recently reviewed yet is that epigenetic modifications may mediate the impact of traumatization of parents to be passed to their offspring. Evidence from epigenetic cell and animal studies has spurred this understanding. In addition, initial recent studies in humans support these notions. The best studied of all the epigenetic mechanisms is the DNA methylation mechanism.

Since trauma and PTSD models in animals may not be well characterized (and it is not very clear how much of the animal knowledge could translate to humans), we find that in this case human studies could provide valuable insight in this area. Despite the limited literature to date in humans, a review of the transgenerational heritability as it pertains to trauma will provide both clinicians and researchers with an overview on where we stand and a clearer vision of where we should go in terms of future research directions. Transgenerational epigenetic transmission is defined as the transmission of genomic information (in this paper DNA methylation) from one generation to the next without changing the main structure of DNA (i.e., nucleotides sequence).

Also, it should be noted that there are 2 subtypes of studies reviewed below, which involve transgenerational effects of DNA methylation changes. In the first one, the samples of mothers were pregnant during the time of trauma. While the second one, the mothers were not pregnant, and the trauma occurred before pregnancy. It seems for the latter, methylation changes to the mothers’ DNA is a necessity for the transmission to occur transgenerationally. On the other hand, when trauma occurs during pregnancy, (through methylation changes), the changes in the mothers are (in theory) not a necessity for the changes in the offspring to occur. While the mechanism and intensity of transgenerational genomic transmission (among other factors) may differ between these types of studies, the field has yet to clarify these differences in future research.

It is still unclear whether trauma-induced epigenetic modifications can be passed from traumatized individuals to subsequent generations of offspring. Thus, the key question of this paper is: whether trauma-induced DNA methylation modifications can be passed from traumatized individuals to subsequent generations of offspring. The aim of this paper is thus to review this emerging literature on the transgenerational effects of trauma on DNA methylation in humans and to provide insights from the current literature (PubMed was reviewed from inception until March 2018), then end with recommendations for future research directions.

## 2. Studies on Transgenerational Effect of Trauma and PTSD

Perroud et al. [14] examined the impact of the Tutsi genocide on the children of the women who were pregnant while genocide was ongoing in Rwanda. In 2011, more than 20% of the Rwandan population met criteria for PTSD [15]. The genocide took place in 1995, the authors investigated if the risk for PTSD had been associated with epigenetic modifications in the children of women who were pregnant at the time of the genocide [14]. Twenty-five widows and their children were included in the study, as well as 25 Rwandan control women who were pregnant at the time, but who were living abroad. Peripheral blood leukocytes were obtained and methylation levels of the promoter regions of the glucocorticoid receptor *NR3C1* and the mineralocorticoid receptor *NR3C2* were examined. Cortisol levels, mineralocorticoid, and glucocorticoid levels were also measured. As expected, both mothers exposed to genocide and their children had significantly higher levels of PTSD and depression than the control group. They also showed higher methylation levels at exon 1F promoter of *NR3C1*, at CpG3-CpG9. Methylation at NR3C2 was not statistically significantly different between the two groups. Mothers and children exposed to trauma had a lower cortisol level than non-exposed mothers and their children (lower cortisol levels were found to be related to PTSD [16]). There was a negative correlation between *NR3C1* methylation and glucocorticoid levels in plasma, but no correlation was found between mineralocorticoid level and NR3C2 methylation level [14]. This interesting study by Perroud et al. highlighted the methylation changes in exposed individuals as well as in their children [14] and suggested that trauma-induced methylation changes in humans can be transmitted from parents to children. Limitations of the study include small sample size and possible confounding of the results. Confounding might occur due to the following reasons in this study: (1) since the 2 groups lived in different countries, unknown confounding variables that might have influenced the comparison with the exposed group might have affected the results; (2) the authors did not mention matching the exposed group versus the control group for possible confounders; (3) other confounding factors, such as health of participants at the time of their pregnancy, parenting style, alcohol consumption, tobacco use, and dietary factors could have played a role.

Yehuda and Daskalakis et al. [17] examined transgenerational methylation changes of Holocaust survivors on FKBP5, a moderator of glucocorticoid activity. The researchers examined 32 Holocaust survivors and their 22 offspring, as well as 8 control subjects and their 9 offspring. Blood samples were obtained for quantification of FKBP5 methylation and cortisol levels. There were significantly higher FKBP5 intron 7 methylation levels in Holocaust survivors, but significantly lower FKBP5 intron 7 methylation levels in their offspring. The authors suggested that this opposite effect seen on FKBP5 intron 7 methylation levels might be attributable to biological accommodation in the offspring. More research is needed to replicate these findings in a larger cohort. Limitations of the study include its limited sample size, as well as, the presence of other factors which are impossible to control for in the population, such as the extreme starvation conditions of the Holocaust survivors, which could have also contributed to the effect seen. Despite the limitations of this study, it is quite a unique and informative study. If further replicated in a larger study, these findings could change the way our field conceptualizes trauma and PTSD.

## 3. Transgenerational Effects of Trauma and Stress and Physical Health

Although the main focus of the review is on the psychological and psychiatric effect of trauma transgenerationally via methylation changes, another area that could be impacted by trauma and stress (and supported by emerging and preliminary evidence) is that the transgenerational effects may not only affect psychological health but also physical health [18]. These physical effects are briefly demonstrated in the following studies.

A study by Mulligan et al. (2012) tested if prenatal maternal stress in the Congo population resulted in glucocorticoid receptor *NR3C1* methylation changes in the offspring (that may lead to increasing risk of adult-onset disease) [19]. The researchers found that in 25 mother-newborn dyads, a significant correlation between prenatal maternal stress and newborn increased methylation levels in the promoter of the glucocorticoid receptor gene (*NR3C1*). Maternal stress was also correlated with low birth weight. They suggested that increased methylation may “constrain plasticity in subsequent gene expression and restrict the range of stress adaptation responses possible in affected individuals”, thus increasing the risk for chronic diseases later in life.

For birth weight, they found that among the stressors examined, war stress has the strongest correlation, as well as, the largest effect size, accounting for 35% of the variance in birth weight (Pearson’s correlation = −0.62, *p* = 0.0009). Within all war stress variables, rape accounted for 31% of birth weight variance “and eclipses the effect of other war stressors”.

This increased methylation of the *NR3C1* gene could increase stress reactivity and could have a long-lasting effect on vulnerability to stress and trauma and chronic disease development.

A study by Radtke al (2011) examined the methylation of 10 CpG sites in the promoter region of the *NR3C1* gene in mothers who suffered pre-pregnancy or during pregnancy exposure to intimate partner violence (IPV) and their children at the age of 10–19 [20]. According to the researchers, this was the first study to examine the effect of the trauma (IPV) over the long term (i.e., at offspring age of 10–19 years). The IPV was assessed retrospectively using the composite abuse scale (CAS) [21]. The study showed methylation of the *NR3C1* promotor gene of the children is influenced by their mother’s experience of IPV during pregnancy. There was no association between the mother’s *NR3C1* methylation and IPV. The authors concluded that as “these sustained epigenetic modifications are established in utero, we consider this to be a plausible mechanism by which prenatal stress may program adult psychosocial function”.

However, we would like to note that the relationship could or could not be causative, as this study only showed an association, and thus causation could not be inferred based only on this study. Other limitations of this study include: the limited sample size of 25 and the retrospective recall of IPV with the possibility (though unlikely) of inaccurate recall. However, this finding if replicated would be important, as this is the first study to show the persistence (over many years) of trauma associated methylation of the regulator gene of the HPA-axis in the offspring. Moreover, a general caution is that, thought the reviewed studies in this section did not actually address the long-term physical effects directly, those are assumed based on results. For instance, the Mulligan et al. (2012) study only examined birth weight, which is not a long-term effect [19]. However, low birth weight has certainly been linked to chronic health effects later in life (as has been shown in several other studies) [22,23].

## 4. Discussion

The limited available literature in humans suggests that children of parents who had suffered from extreme trauma have methylation modifications associated with trauma and PTSD. This was also true in studies done in different populations, as in case of Yehuda and Daskalakis et al. [17] and Perroud et al. [14]. This may support the combined influence, not only of environmental trauma, but also of the biological component of PTSD risk. Moreover, this PTSD risk can be passed from generation to generation. Of interest, the environmental transgenerational effects that lead to change in DNA methylation in the offspring has also been demonstrated in several animal models. For instance, dietary supplements during pregnancy was associated with increased methylation (of the Agouti coat color gene) and causes permanent change in coat color [24,25] thus, suggesting that prenatal and natal environmental interventions could induce epigenetic alterations with robust impact on stress-related disorders. These environmental interventions could range from trauma and stress to dietary and pharmacological interventions [26].

We want to point out, though, that in the Yehuda and Daskalakis et al. [17] study, the DNA methylation seen in the offspring was noted to be opposite from that seen in their parents. Although this is a perplexing finding, one possibility is that it may be a compensatory mechanism, as has been seen in Holocaust survivors who suffered extreme starvation (which is another environmental factor that can affect the epigenome) and whose offspring were more prone to metabolic syndrome [27].

However, in line with the well-established notion that glucocorticoids are stress hormones, many of the studies reviewed found that the glucocorticoid receptor (*NR3C1*) gene is associated with methylation changes. For instance, maternal exposure to intimate partner violence during pregnancy was associated with increased *NR3C1* DNA methylation in teenage children [20]. Maternal exposure to war violence or rape during pregnancy was associated with increased methylation in the *NR3C1* promoter region in newborns [19,28].

Some weaknesses of this review and the current literature include that there are limited number of studies, and most had small sample size. It is also unknown if there was publication bias related to studies on this topic, where studies with negative results were not published. Also, there may have been other confounding factors (which could have led to epigenetic changes) that were not accounted for by the studies. Moreover, several of the studies represent cross-sectional studies, where causation cannot be inferred directly from these studies. 

It should be noted that these findings are still preliminary and should be taken with caution. Thus, the findings need to be replicated in larger studies with a control group, which will also increase our understanding and further specify and solidify these findings. Future studies should also (as some of the studies reviewed did) clarify the timing of the trauma: of whether it occurred prior to pregnancy or during pregnancy. In addition, further differences based on the trimester of pregnancy may be important to know. Also, the chronicity versus acuity of the transgenerational trauma needs to be considered and studied. Future research of this topic may lead to identification of biomarkers of trauma and PTSD risk and to greater advances into the prevention and treatment of PTSD. In addition, future studies should also follow the research recommendations of the National Advisory Mental Health Council Workgroup on Genomics (NAMHC), including using strict standards of statistical rigor for disease association, moving away from the candidate gene approaches, which have not been very fruitful, and moving towards well-powered genetic and epigenetic association studies [29,30]. Expanding the genetic association studies beyond the DSM nosology, for instance studying the effect of trauma per se on and the methylation changes that associate with the development of psychopathology in general (and not necessary PTSD alone), would be helpful (since heritability is shared across psychiatric disorders). Replicating the finding in diverse populations and the differences that may be unique to a particular population would be helpful. This will require data sharing between many centers to generate enough sample size and meaningful strong significances in genome-wide studies as indicated also by the NAMHC [30].

If the findings in the reviewed studies are confirmed definitively and further specified, it can both inform clinical research, as well as prove clinically beneficial in predicting risk to PTSD development as well as treatment planning.

## 5. Conclusions

Despite the limitation of the current literature, there seems to be accumulating evidence to suggest the transgenerational transmission of DNA methylation changes from parents to children. This area merits further replication of the presented findings. In addition, as some of the studies reviewed did, future studies should also clarify the timing of the trauma (of whether it occurred prior to pregnancy or during pregnancy), and if there are further differences based on the trimester of pregnancy. Also, the chronicity versus acuity of the transgenerational trauma needs to be considered and studied. Future research of this topic may lead to identification of biomarkers of trauma and PTSD risk and to greater advances in the prevention and treatment of PTSD.

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
