# Peer review of "The Effects of Trauma, with or without PTSD, on the Transgenerational DNA Methylation Alterations in Human Offsprings"

_brainsci, 2018, doi:10.3390/brainsci8050083_

Reviewer 1 Report

The topic reviewed here is very timely and important – whether there are transgenerational epigenetic effects from PTSD. However, I have reservations about this review article. It is based essentially on 4 studies, all of which are small (which I can understand, given the difficulty of studying this question). I just think it might be a bit too early to be writing a review paper on this topic, since review papers are useful to scientists when there is a large body of literature that must be digested in a concise way. In this case, someone who wants to review the literature on this topic could as well read the four studies, and it wouldn’t consume too much time. I recognize that this topic is very challenging to study and also very needed, but probably the field should develop more before a review paper is written. The authors may consider converting this paper into a commentary about this topic. I enjoyed their perspectives and they seemed to be very well versed in the subject matter.

Some detailed comments about the paper itself:

-          There are a few grammatical errors, so I think proofing the paper would be useful. For example, line 95 should read “humans” not “human”; line 194 should read “in DNA methylation” not “to DNA methylation”; line 210, drop the word “the” before “stress”; line 123 could consider changing to “(those in Rwanda during the genocide)”, thus deleting “who was”; line 176 insert “the” between “that” and “negative”;

-          I did not see a statement in the paper about the date up to which the review was conducted. This would be helpful.

-          For clarity, I would change the term “intrauterine” in 162 to “during pregnancy.”

-          It is unclear what the authors mean in line 172 as “an association study.” Do they mean a cross sectional study? This was not explained, so the reader does not understand the study design of the cited study.

-          The studies described in the last paragraph of section 3 (lines 176-186) do not appear to be transgenerational studies but studies of early life exposure and later life health outcomes (refs 24-31). As such, they should not be included here, since they don’t involve offspring.

-          On line 211, there seems to be a typo. It should read “intimate partner” not “interpartner.”

-          There is never an explanation of the implications of increased methylation in NR3C1, which was a result of a few of the studies presented. This would be an important part of the discussion.

-          The authors mention that they will end with recommendations for future research directives. This is only mentioned for the first time in the Conclusions. These recommendations should be mentioned in the Discussion and then re-iterated in the Conclusions. However, the research is yet so sparse that there are only a few, listed recommendations, which aren’t expanded upon much (i.e., further replication, specifying regions of methylation (but the studies here did specify the regions of methylation), whether trauma occurred prior to pregnancy or during (again, these were specified in the reviewed studies), if differences are based on trimester, and chronicity vs. acuity of transgenerational trauma (a very good suggestion)).

Author Response

The topic reviewed here is very timely and important – whether there are transgenerational epigenetic effects from PTSD. However, I have reservations about this review article. It is based essentially on 4 studies, all of which are small (which I can understand, given the difficulty of studying this question). I just think it might be a bit too early to be writing a review paper on this topic, since review papers are useful to scientists when there is a large body of literature that must be digested in a concise way. In this case, someone who wants to review the literature on this topic could as well read the four studies, and it wouldn’t consume too much time. I recognize that this topic is very challenging to study and also very needed, but probably the field should develop more before a review paper is written. The authors may consider converting this paper into a commentary about this topic. I enjoyed their perspectives and they seemed to be very well versed in the subject matter.

We have cutout some of the less important details in the manuscript. We appreciate the reviewer complement that he/she “enjoyed their perspectives and they seemed to be very well versed in the subject matter.”

Some detailed comments about the paper itself:

-          There are a few grammatical errors, so I think proofing the paper would be useful. For example, line 95 should read “humans” not “human”; line 194 should read “in DNA methylation” not “to DNA methylation”; line 210, drop the word “the” before “stress”; line 123 could consider changing to “(those in Rwanda during the genocide)”, thus deleting “who was”; line 176 insert “the” between “that” and “negative”;

Thank you. We have now done these changes.

-          I did not see a statement in the paper about the date up to which the review was conducted. This would be helpful.

Added now.

-          For clarity, I would change the term “intrauterine” in 162 to “during pregnancy.”

Done

-          It is unclear what the authors mean in line 172 as “an association study.” Do they mean a cross sectional study? This was not explained, so the reader does not understand the study design of the cited study.

This is now clarified.

-          The studies described in the last paragraph of section 3 (lines 176-186) do not appear to be transgenerational studies but studies of early life exposure and later life health outcomes (refs 24-31). As such, they should not be included here, since they don’t involve offspring.

Thank you. We deleted this last paragraph of section 3 from the results, as suggested by the reviewer, as it is only part of the discussion and moved it to the discussion section, where it fits better.

-          On line 211, there seems to be a typo. It should read “intimate partner” not “interpartner.”

Done

-          There is never an explanation of the implications of increased methylation in NR3C1, which was a result of a few of the studies presented. This would be an important part of the discussion.

We now provided discussion of the implication of increased methylation in NR3C1 in the Discussion section as can be gleaned by the literature in the field.

-          The authors mention that they will end with recommendations for future research directives. This is only mentioned for the first time in the Conclusions. These recommendations should be mentioned in the Discussion and then re-iterated in the Conclusions. However, the research is yet so sparse that there are only a few, listed recommendations, which aren’t expanded upon much (i.e., further replication, specifying regions of methylation (but the studies here did specify the regions of methylation), whether trauma occurred prior to pregnancy or during (again, these were specified in the reviewed studies), if differences are based on trimester, and chronicity vs. acuity of transgenerational trauma (a very good suggestion)).

We now added recommendations to the Discussion sections and deleted the redundancy and unnecessary material in the Conclusion, as suggested by the reviewer.

Reviewer 2 Report

Manuscript ID: brainsci-269104

The effects of trauma, with or without PTSD, on the transgenerational epigenetic DNA methylation in the Human Offsprings

Thank you for the opportunity to review this manuscript. This paper represents a review of the current literature on transgenerational epigenetic changes as a function of trauma exposure, in human populations. While this review adds to the literature by thoughtfully summarizing existing literature in this growing area and offers a useful critique of current limitations in the field, there were a number of overarching and more specific concerns that need to be addressed, in order to increase clarity and impact of this work. These are detailed below; it is hoped that these comments will be useful in future revisions.  

Introduction:

The introduction would benefit from some reorganization--specifically, there are some sentences/sections that feel out of place and distract from the main points being made (detailed below in Minor points). I would also recommend perhaps changing the use of the term “epigenetics” in many places, as this is a very broad term encompassing many different types of changes, and DNA methylation is the most commonly studied form of epigenetic alteration and seems to be much of what was examined in the reviewed studies. For example, in line 64 (p. 2) it reads odd to describe “an association between depression and epigenetics” and “suicide and epigenetics” as well as then describing “hypermethylation of BDNF”—it seems that it was methylation changes associated with depression and suicide, at specific locations perhaps. Would be useful to edit for clarity. Additionally, further description of methylation, following description of epigenetic mechanisms would also be useful for this review paper given the above point regarding its frequency in the literature.

Studies on Transgenerational effect of trauma and PTSD:

This section includes a useful review and summary of existing studies. Prior to description of extant work, however, it would be useful if the concept of transgenerational effects were clearly operationalized early on in the review. It would also be helpful to very clearly describe the differences between studies with samples of mothers who were pregnant during the time of trauma and those that were not, as this has different interpretations and implications on epigenetic alternations noted.

Discussion

The discussion of limitations was useful, but this section would also benefit from a re-organization, that clearly separates discussion of limitations of extant work and also summarizes findings to date that point to future directions.

Minor points.

1.     The title is in need of some rewriting for typographical/grammatical issues (should be PTSD and Offspring) as well as for clarity: “epigenetic DNA methylation” is confusing and a bit redundant. Perhaps either “epigenetic modifications” or “DNA methylation changes” only.

2.     Some minor grammatical points in the abstract (line 20 “consequent” and line 27 “to be passed transgenerational”)

3.     It is unclear what the paragraph on page 2, lines 45-51, adds and feels a bit off point.

4.     I would suggest edits to p. 2 line 54-55. As it reads, it appears to be saying “structural genetic-up AND heritability" are estimated at 30-70%. This is confusing and not accurate. Heritability has been estimated at 30-70%, which is a different point than that being made with regard to structural genetic make-up (i.e., molecular genetic work, I would assume—both candidate gene and GWAS methodologies).

5.     The text in lines 70-76 (p. 2) don’t seem to fit with the points trying to be made—was a bit confusing to read—perhaps remove.

6.     In section 3, the organization/separation of the paragraphs was difficult to follow; editing and reorganization is recommended. 

Author Response

The effects of trauma, with or without PTSD, on the transgenerational epigenetic DNA methylation in the Human Offsprings

Thank you for the opportunity to review this manuscript. This paper represents a review of the current literature on transgenerational epigenetic changes as a function of trauma exposure, in human populations. While this review adds to the literature by thoughtfully summarizing existing literature in this growing area and offers a useful critique of current limitations in the field, there were a number of overarching and more specific concerns that need to be addressed, in order to increase clarity and impact of this work. These are detailed below; it is hoped that these comments will be useful in future revisions.  

Thank you very much for the reviewer’s positive evaluation and are glad that the review “adds to the literature by thoughtfully summarizing existing literature in this growing area and offers a useful critique of current limitations in the field.” Below we address the concerns and clarification of these issues.

Introduction:

The introduction would benefit from some reorganization--specifically, there are some sentences/sections that feel out of place and distract from the main points being made (detailed below in Minor points). I would also recommend perhap s changing the use of the term “epigenetics” in many places, as this is a very broad term encompassing many different types of changes, and DNA methylation is the most commonly studied form of epigenetic alteration and seems to be much of what was examined in the reviewed studies. For example, in line 64 (p. 2) it reads odd to describe “an association between depression and epigenetics” and “suicide and epigenetics” as well as then describing “hypermethylation of BDNF”—it seems that it was methylation changes associated with depression and suicide, at specific locations perhaps. Would be useful to edit for clarity. Additionally, further description of methylation, following description of epigenetic mechanisms would also be useful for this review paper given the above point regarding its frequency in the literature.

As the reviewer suggested, we now removed the distracting sentences and have changed the use of the word epigenetics to “DNA methylation” in most parts when appropriate (except where we describe the definition of epigenetics and the like). We have also defined and described DNA methylation, as suggested. We also changed the sentence to “hypermethylation of BDNF” to “hypermethylation of BDNF promotor” and rephrased association of depression and suicide with “DNA methylation changes” as elaborated in the cited reference.

Studies on Transgenerational effect of trauma and PTSD:

This section includes a useful review and summary of existing studies. Prior to description of extant work, however, it would be useful if the concept of transgenerational effects were clearly operationalized early on in the review. It would also be helpful to very clearly describe the differences between studies with samples of mothers who were pregnant during the time of trauma and those that were not, as this has different interpretations and implications on epigenetic alternations noted.

As suggested, we defined/operationalized the concept of transgenerational effects in this paper. We also added a brief note clarifying that 2 types of transgenerational studies that will be reviewed (trauma during pregnancy and those with trauma before pregnancy).

Discussion

The discussion of limitations was useful, but this section would also benefit from a re-organization, that clearly separates discussion of limitations of extant work and also summarizes findings to date that point to future directions.

Minor points.

1.     The title is in need of some rewriting for typographical/grammatical issues (should be PTSD and Offspring) as well as for clarity: “epigenetic DNA methylation” is confusing and a bit redundant. Perhaps either “epigenetic modifications” or “DNA methylation changes” only.

Changed as suggested.

2.     Some minor grammatical points in the abstract (line 20 “consequent” and line 27 “to be passed transgenerational”)

Done

3.     It is unclear what the paragraph on page 2, lines 45-51, adds and feels a bit off point.

We now deleted this paragraph

4.     I would suggest edits to p. 2 line 54-55. As it reads, it appears to be saying “structural genetic-up AND heritability" are estimated at 30-70%. This is confusing and not accurate. Heritability has been estimated at 30-70%, which is a different point than that being made with regard to structural genetic make-up (i.e., molecular genetic work, I would assume—both candidate gene and GWAS methodologies).

Changes and clarified as suggested.

5.     The text in lines 70-76 (p. 2) don’t seem to fit with the points trying to be made—was a bit confusing to read—perhaps remove.

Deleted this part, as suggested.

6.     In section 3, the organization/separation of the paragraphs was difficult to follow; editing and reorganization is recommended. 

We organized and clarified this section.

Round  2

Reviewer 2 Report

Manuscript ID: brainsci-269104 (Revision)

The authors were responsive to reviewer comments overall, and it is noted that the additional text added was useful and strengthened the paper. However, the main concern is that throughout the manuscript, these additions, both large sections of text and small additions, need to be incorporated within the manuscript as a whole. As it currently stands, the sections were simply added in without adjusting surrounding text to ensure flow and clarity and prevent repetition. The added sections also have a number of grammatical and typographical errors and require proofreading. Further, other concerns with the manuscript remain, detailed below. I will also note that Reviewer 1’s suggestion of shifting to a commentary as compared to a review seems to be a useful approach that might be worth considering.

My other, remaining suggestions are as follows:

1.     There remains a continued need to further organize the introduction to clearly set up what has been previously reviewed in the related literature and specifically what the present review is covering and why.

2.     The paragraphs throughout are confusing and separated in inconsistent ways. For example, in section three, the same study is described in multiple paragraphs, one of which is only a sentence long.

3.     I would be careful in section 3 with discussing “physical long-term effects” because most of the reviewed studies don’t actually address long-term physical effects—those are assumed/implied based on results, but not tested. The Mulligan paper,  for example, is only examining birth weight, which is not a long term effect. While low birth weight has certainly been linked to chronic health effects later in life, citations to that effect need to be included, as the study itself can’t speak to that point.

4.     Line 192 appropriately mentions the fact that causation cannot be inferred on a single cross-sectional study. I would recommend that this point be further included in the overall discussion/limitation of methylation studies, as reverse causation is a broader concern of all existing methylation work (I would assume this is also the case for transgenerational changes as well). Especially when measuring methylation in individuals years after trauma exposure (in utero or not) it is still a strong possibility that differences are a function of current disease states as opposed to maternal transmission of these changes.

5.     With regard to the text that was moved to the discussion section (line 226), as the other reviewer pointed out, discussion of early life exposure and later health outcomes are tangential to the review topic, even in the discussion section. Early life trauma and negative outcomes are not getting at transgenerational risk (they are not examination of offspring of trauma-exposed mothers). This should be removed.

6.     Minor point: it looks like, based on the comment bubbles, that the new references were not incorporated into the text and I am unsure if they were added to the reference list. 

Author Response

This letter refers to the revision of our manuscript entitled “The Effects of Trauma, with or without PTSD, on the Transgenerational DNA Methylation alterations in Human Off springs”. We appreciate the time and efforts by the editor and reviewers in reviewing this manuscript. We have addressed all issues indicated in the review report, and hope that the revised version can meet the journal publication requirements. We believe that the paper has been much improved as a result of the revision. I attach a clean version of the manuscript (and another which includes changes indicated with “tracked changes”).

Yours sincerely,

Nagy Youssef

Nagy Youssef, MD

Associate Professor

Director, Mood and Trauma clinic

Department of Psychiatry and Health Behavior

And Office of Academic Affairs

Medical College of Georgia at Augusta University

997 St. Sebastian Way, Augusta, GA 30912

Tel: 706.721.6963

Fax: 706.434.3200

nyoussef@augusta.edu

Comments from the editors and reviewers and responses:

The authors were responsive to reviewer comments overall, and it is noted that the additional text added was useful and strengthened the paper. However, the main concern is that throughout the manuscript, these additions, both large sections of text and small additions, need to be incorporated within the manuscript as a whole. As it currently stands, the sections were simply added in without adjusting surrounding text to ensure flow and clarity and prevent repetition. The added sections also have a number of grammatical and typographical errors and require proofreading. Further, other concerns with the manuscript remain, detailed below. I will also note that Reviewer 1’s suggestion of shifting to a commentary as compared to a review seems to be a useful approach that might be worth considering.

Response: We thank the reviewer for the comment that “authors were responsive to reviewer comments overall, and it is noted that the additional text added was useful and strengthened the paper.” We now improved and corrected the flow for be better incorporation within the paper and corrected any grammatical and typographical errors found.

My other, remaining suggestions are as follows:

1.     There remains a continued need to further organize the introduction to clearly set up what has been previously reviewed in the related literature and specifically what the present review is covering and why.

Response: We organized the introduction and further clarified what this review is covering and why.

2.     The paragraphs throughout are confusing and separated in inconsistent ways. For example, in section three, the same study is described in multiple paragraphs, one of which is only a sentence long.

Response: We corrected confusion and deleted redundancies in section three.

3.     I would be careful in section 3 with discussing “physical long-term effects” because most of the reviewed studies don’t actually address long-term physical effects—those are assumed/implied based on results, but not tested. The Mulligan paper, for example, is only examining birth weight, which is not a long term effect. While low birth weight has certainly been linked to chronic health effects later in life, citations to that effect need to be included, as the study itself can’t speak to that point.

Response: We clarified this now and deleted the word “long-term,” and clarified that long-term effect was not tested directly and used the reviewer’s suggestion of paraphrasing as written. We added citation to the indirect long-term health effects of low birth weight, as suggested by the reviewer.

4.     Line 192 appropriately mentions the fact that causation cannot be inferred on a single cross-sectional study. I would recommend that this point be further included in the overall discussion/limitation of methylation studies, as reverse causation is a broader concern of all existing methylation work (I would assume this is also the case for transgenerational changes as well). Especially when measuring methylation in individuals years after trauma exposure (in utero or not) it is still a strong possibility that differences are a function of current disease states as opposed to maternal transmission of these changes.

Response: We now added to the limitation section that causation cannot be inferred from cross-sectional studies, as suggested by the reviewer.

5.     With regard to the text that was moved to the discussion section (line 226), as the other reviewer pointed out, discussion of early life exposure and later health outcomes are tangential to the review topic, even in the discussion section. Early life trauma and negative outcomes are not getting at transgenerational risk (they are not examination of offspring of trauma-exposed mothers). This should be removed.

Response: We now removed this part from the discussion section, as suggested.

6.     Minor point: it looks like, based on the comment bubbles, that the new references were not incorporated into the text and I am unsure if they were added to the reference list.

Response: The added references are now incorporated in the text (and they are also in the reference list).
